# Predicting Return to Work after Head and Neck Cancer Treatment Is Challenging Due to Factors That Affect Work Ability

**DOI:** 10.3390/cancers15194705

**Published:** 2023-09-25

**Authors:** Ylva Tiblom Ehrsson, Marta A. Kisiel, Yukai Yang, Göran Laurell

**Affiliations:** 1Department of Surgical Sciences, Section of Otorhinolaryngology and Head & Neck Surgery, Uppsala University, 751 85 Uppsala, Sweden; goran.laurell@uu.se; 2Department of Medical Sciences, Occupational and Environmental Medicine, Uppsala University, 751 85 Uppsala, Sweden; marta.kisiel@medsci.uu.se; 3Department of Statistics, Uppsala University, 751 20 Uppsala, Sweden; yukai.yang@statistik.uu.se

**Keywords:** clinical factors, head and neck cancer, return to work, sociodemographic factors, work-related factors

## Abstract

**Simple Summary:**

This prospective study examined the impact of various factors on return to work (RTW) for head and neck cancer (HNC) survivors at 3 and 12 months after treatment. The study included 227 participants aged 65 years or younger. Within 3 months, 92 participants RTW and 30 retired. At 12 months, 80 participants were still working and another 51 participants had RTW. Hindrances to RTW at 3 months were advanced tumour stage (stage III and IV), whereas hindrances at 12 months were oral and larynx cancer. Facilitators of RTW included having a white-collar job, living in a relationship and early tumour stages. The study underscored that disease stage significantly hindered RTW, and work type and having a spouse or partner were nonclinical factors influencing RTW. These findings offer valuable insights for healthcare professionals and policymakers, which can aid in the development of strategies and support systems to enhance the RTW experience for HNC survivors.

**Abstract:**

Striving to return to work is of great importance to many cancer survivors. The purpose of the study is to prospectively investigate the factors that hinder and facilitate return to work (RTW) at 3 and 12 months after the end of treatment in head and neck cancer (HNC) survivors and whether these factors influence the ability to continue working after treatment. Participants (*n* = 227) aged ≤ 65 years at diagnosis with HNC were included. Data were collected before the start of treatment and at 3 and 12 months after the end of treatment. The Rubin causal model was used for statistical analysis. Within the 3-month follow-up period, 92 participants had RTW and 30 had retired. At the 12-month follow-up, 80 of these participants were still working, another 51 participants had RTW, and five participants working still suffered from cancer. The hindrance to RTW within 3 months was advanced tumour stage (stage III and IV) (*p* = 0.0038). Hindrances to RTW at the 12-month follow-up were oral cancer (*p* = 0.0210) and larynx cancer (*p* = 0.0041), and facilitators were living in a relationship (*p* = 0.0445) and a white-collar job (*p* = 0.00267). Participants with early tumour stage (stage I and II) (*p* = 0.0019) and a white-collar job (*p* = 0.0185) had earlier RTW. The conclusion is that disease factors were the most important hindrances to RTW, and type of work and living with a spouse or partner were nonclinical factors influencing RTW.

## 1. Introduction

Striving to return to everyday life, including work, is of great importance to many cancer survivors, and the success rate is traditionally regarded as dependent on the site of the tumour, tumour stage, treatment, and comorbidities [1,2,3].

Head and neck cancer (HNC) includes a group of malignancies located at different sites in the upper aerodigestive tract, and squamous cell carcinoma (SCC) accounts for more than 90%. Treatment for HNC consists of single or combined modality treatment with surgery, radiotherapy, and medical treatment (such as chemotherapy and targeted therapy), which are included in the therapeutic arsenal [4]. Earlier, this cancer type predominantly affected older people who were heavy smokers or suffered from alcohol abuse. However, in recent decades, the demographics have changed due to the increasing incidence of human papillomavirus (HPV)-positive oropharyngeal cancer, which majorly affects younger patients [5]. The ability to return to work (RTW) and continue working is linked to several medical, physical, psychosocial, and social factors experienced by patients [6]. Treatment-related acute toxicities and sequelae can result in early, late, and chronic problems for the cancer survivor [7]. Work-related and sociodemographic factors are also considered to play an important role in the effort to RTW after sick leave in connection with a long period of treatment for HNC [8,9].

There is increasing awareness that RTW after cancer treatment can have important benefits for cancer survivors’ well-being. Moreover, altered treatment regimens and the introduction of rehabilitation interventions suggest the need for further studies on HNC survivors [10]. Only a few comparison studies on RTW exist in the literature; however, a recent review of 29 articles show considerable variability in RTW among HNC survivors [3]. With the increasing incidence of HNC in young working populations, there is a need for more prospective cohort studies to evaluate the risk/beneficial factors that impact RTW [11]. In an intervention review of patients with different cancer diagnoses, RTW strategies were analysed. Low-quality evidence was demonstrated for a similar rate of RTW for psycho-educational interventions and standard care. However, moderate-quality evidence indicated that multidisciplinary interventions including physical, psycho-educational, or vocational components resulted in higher RTW rates than standard care [12]. The purpose of this study was, hence, to prospectively investigate factors that hinder and facilitate RTW 3 and 12 months after the end of treatment in HNC survivors and to determine whether these factors influence the ability to continue working after treatment.

## 2. Materials and Methods

### 2.1. Study Design and Population

This is an ongoing multicentre prospective observational study of patients with HNC registered at https://www.ClinicalTrials.gov (identifier NCT03343236). Two hundred and twenty-seven participants aged 65 years or younger at the time of diagnosis were included in this study from October 2015 to August 2021. Inclusion criteria were age above 18 years, curable untreated HNC, and a performance status of 0–2 according to the Eastern Cooperative Oncology Group Performance Status/World Health Organization Performance Status (WHO PS) [13]. The exclusion criteria were malignant neoplasms previously treated within the last five years (except for skin cancer), inability to understand Swedish language, severe alcohol abuse, and cognitive impairments.

### 2.2. Data Collection

Data were collected on three occasions: before the start of HNC treatment (baseline) and 3 and 12 months after the end of treatment. The rationale for the two follow-ups used in the present study is as follows: at 3 months post-treatment, the first assessment is made of the treatment effectiveness in general, i.e., whether the treatment has cured the cancer patient or not. At 1-year post-treatment, the sick benefit dramatically decreases in Sweden which has an important impact on the patient’s and family’s economic situation. The data are stored in a database (data.dynareg.se). This database was developed to facilitate easy, reliable, and safe data collection for prospective multicentre observational studies.

At the baseline, clinical characteristics and sociodemographic data were collected by a research nurse who collected information from the participants by asking them about their age, sex, marital status, type of accommodation, educational level, smoking status, current working status, and occupation. Additional information was collected from the participants’ medical records, including cancer site, and tumour stage according to the Union for International Cancer Control (UICC)’s 8 staging system, and treatment.

The timing of RTW was tracked at 3 and 12 months after the end of treatment. The participants reported their working status, and from the medical record, the outcome of treatment (cancer-free/recurrence or mortality from cancer/other diseases) was collected. Two categories of RTW were studied: early RTW (0–3 months after the end of treatment) and late RTW (3–12 months after the end of treatment).

The participants were classified into three occupational groups by the first and second authors: (1) white-collar workers, defined as people performing professional, desk manager, and administrative work; (2) blue-collar workers, defined as people performing manual labour including machine operators, assemblers, and occupations with demand for elementary education [14]; and (3) pink-collar workers, defined as healthcare workers (including physicians, nurses, assistant nurses, occupational and physiotherapists, psychologists, and residential workers [15]. The classification of workers was based on the Swedish standard system for the classification of occupations [16].

The study was performed in accordance with the principles of the Declaration of Helsinki of the World Medical Association [17], and approval was granted by the Regional Ethical Review Board in Uppsala, Sweden, No. 2014/447. Informed oral and written consent was obtained from all participants included in the study.

### 2.3. Statistical Analysis

Descriptive data for the continuous variable are presented as mean ± standard deviation (SD), and the categorical variables are presented as numbers (%). Pearson’s chi-squared test was used to analyse cancer stage and RTW. Data were analysed using the statistical software IBM SPSS version 28.0 (IBM, Armonk, NY, USA) and R version 4 (R Foundation for Statistical Computing, Vienna, Austria).

The Rubin causal model [18,19,20,21,22], which conceptualises causal inference in terms of potential outcomes, was used to analyse RTW. The rationale for using the causal model and potential outcomes framework in this study instead of common regression methods, which likely reach the same findings, is that the latter can provide valuable insights and associations between variables, but they are limited in their ability to establish causality, whereas the potential outcomes framework aims to understand the true impact of a particular action or intervention on an outcome of interest while accounting for potential confounding factors. Regarding the research purpose of this paper, the average causal effect (ACE) and potential outcomes framework is a more natural and straightforward statistical analysis.

A good starting point is to consider the question “what causes the patient to RTW after being treated for HNC?” In the context of statistical causal effect analysis, an action variable W can be regarded as a potential cause of a certain outcome. Each observation in the dataset belongs to either the action group (W = 1) whose members were exposed to the action variable, or the control group (W = 0) whose members were not exposed to the action variable. Note that the action group in our study is sometimes termed the treatment group in other fields.

Supposing that there are two random potential outcomes, *Y_i_*(1) and *Y*_i_(0) for an individual (patient) *i*, corresponding to the action W*i* = 1 or 0. The observed outcome is a mixture of two potential outcomes in the following manner: *Y_i_*^obs^ = *Y_i_*(0)(1 − *W*_i_) + *Y_i_*(1) *W*_i_. If *Y_i_*(1) and *Y_i_*(0) have different expectations, we say that the action *W*_i_ has a causal effect. The estimate of interest under the causal inference context is then the ACE defined to be (ACE = E[*Y_i_*(1) − *Y_i_*(0)]) where the average is over the population of the individuals. Both *Y_i_*(1) and *Y_i_*(0) are random because of the varying nature of the covariates and the latent error. ACE is the effect size of interest in this study, as it represents the average difference between two potential outcomes over the population; therefore, it indicates the causal effect or the action truly causes the parameter of interest if it is nonzero. The two terms “ACEe” and “ACEp” will be used in the following to represent the estimate of ACE and the corresponding *p*-value, respectively.

Ideally, if we can observe both the two outcomes *Y_i_*(1) and *Y_i_*(0) for the same patient, then we can simply take the subtraction of them and average across the patients to estimate the causal effect. However, in practice, the counterfactual can never be observed. This is the “missing observation” problem in causal effect analysis.

We hope that the two groups in the study should be well balanced in the sense that, if there is one patient in one group, there should be a similar one (in terms of covariates) in another group so that we can still observe the “counterfactuals” to some extent. However, for an observational study of this kind, we cannot guarantee that the assignment of action exposure is purely random or that the action and control groups in the original data set were well balanced. The assignment of action exposure (for some variables) in practice may be based on the patient’s characteristics, which causes the confounding problem that produces a bias in the estimation of ACE.

In order to reduce the bias caused by any potential confounding, we applied the propensity score matching (PSM) technique to balance the covariates and alleviate potential bias in the estimation. The “propensity score“ evaluates how likely a certain patient is to have been exposed to either of the two groups based on the observational data. The goal to perform the matching is to assign patients with similar propensity scores in both the action and control groups so that the two groups are more comparable.

We employed the logistic regression method to estimate the propensity scores for each observation, as it is the most common and sophisticated method for PSM estimation. For details on the propensity score matching technique, see Rosenbaum and Rubin [23].

The variables of interest that we model and investigate in this study are RTW within certain periods (3 and 12 months) after corresponding medical treatments for each individual in the dataset.

#### Data Manipulations

The variables used in the analysis after data manipulation were the dependent variable, covariates, and action variables. The indicator of RTW was used as a common dependent variable in all models.

## 3. Results

### 3.1. Description of Participants before the Start of Treatment

Of the 227 participants with HNC included in this study, 66 were women. The mean age of the participants was 55.1 years (range 22–65 years). The clinical, work-related, and sociodemographic characteristics of the study population are presented in Table 1.

### 3.2. Early Return to Work (RTW)

Within the 3-month follow-up, 92 (41%) participants (age range 27–65 years, mean 53.2 years) had early RTW. Two were unemployed, 99 were on sick leave, 30 were retired, and 4 died. Eighty of the ninety-two participants who had early RTW continued to work at the 12-month follow-up. The remaining 12 participants either retired (*n* = 6), went back on sick leave (*n* = 2), died (*n* = 2), or dropped out of the study (*n* = 2), (Figure 1).

### 3.3. Late Return to Work (RTW)

Between the 3- and 12-month follow-up, another 51 participants RTW. The remaining participants were unemployed (*n* = 7) and were still on sick leave (*n* = 26). Additionally, 7 participants retired and 11 died. At the 12-month follow-up, a total of 131 participants were employed (Figure 1).

### 3.4. The Stage of Cancer and Treatment Outcome

A total of 99 participants were on sick leave and 92 participants were working at the 3-month follow-up. At the 12-month follow-up, the number of participants on sick leave had decreased to 28 and the number of participants working had increased to 131 (Figure 1 and Table 2). A significant difference was observed between participants working and those on sick leave regarding cancer stage at the 3-month follow-up and at the 12-month follow-up (Table 2).

Of the 227 included participants, 11 were non-responders to cancer treatment or had early recurrence within 3 months of treatment completion. One of these 11 participants (diagnosed with cancer of unknown primary, stage IVB) had RTW early (at the 3-month follow-up) but was not working at the late follow-up. One participant was diagnosed with stage II laryngeal cancer with a partial response after RT, underwent transoral laser microsurgery, and was thereafter cancer-free, with an RTW between the 3- and 12-month follow-ups. None of the other non-responders or participants with early recurrence RTW. Before the 3-month follow-up, three participants died. At the 12-month follow-up, two participants with early recurrence had died.

Between the 3- and 12-month follow-ups, 31 participants (24 males and 7 females) were diagnosed with cancer recurrence. The sites of cancer recurrence were the oropharynx in 15 participants, oral cavity in 12 participants, and in four other sites. Of these 31 participants, seven had early RTW and one was unemployed. At the 12-month follow-up, one of the seven participants with early RTW had died, and the other six continued to work, even though four of them were not cancer-free and the other two were cancer-free after salvage surgery. Additionally, two participants with cancer recurrence had RTW (cancer-free), and the unemployed participant had RTW (not cancer-free) between 3 and 12 months. In total, five participants with active cancer disease (two were unsuccessfully treated with salvage surgery) and four participants who were cancer-free after salvage treatment had RTW at 12 months. The remaining participants with cancer recurrence between the 3- and 12-month follow-ups had retired *n* = 7, continued sick leave *n* = 6, were unemployed *n* = 1, or died *n* = 8. Of the 31 participants with cancer recurrence, eight died and had stage III *n* = 4, IVA *n* = 3, and IVB *n* = 1 cancer.

### 3.5. Clinical, Work-Related, and Sociodemographic Factors and Return to Work

Stage III–IV (*p* = 0.0038) hindered early RTW. Furthermore, oral cancer (*p* = 0.0210) and larynx cancer (*p* = 0.0041) were hindrances to late RTW and being a white-collar worker (*p* = 0.0267) or living with a spouse or partner (*p* = 0.0445) were facilitating factors at the 12-month follow-up. No other factors were significantly different (Table 3). In Table 3, the first column is the action variables. The second column (Yes) reports the percentage numbers of RTW (RTW rate) exposed to the action variable, and the parenthesis is the total number of participants in the action group. The third column (No) reports the percentage numbers of RTW not exposed to the action variable, and the parenthesis is the total number of patients in the control group. The fourth column (ACEe) reports the estimates of the ACE. Intuitively, it is supposed to be the difference between the percentages in the second and third columns. However, as we employed the propensity score matching (PSM) technique in this study, the action and control groups have been rebalanced according to the estimated propensity scores. We observe that the estimate of ACE in the second row, for example, is −0.1047 which even has different sign compared to the difference between 53.85% and 44.53%. This also reflects the fact that the original two groups are not balanced and there is confounding. The fifth column (ACEp) reports the corresponding p-values of the estimated ACE. If the estimate is significant, a positive value implies that it facilitates RTW, whereas a negative value implies that it hinders RTW. Excluded participants are those who retired or are deceased.

A comparison between the participants who RTW at some point within 3 months after treatment (*n* = 92) and those of the participants who continued to work at 12 months (here called early RTW and continuing working, *n* = 80) is presented in Table 4. Stage III–IV was a hindering factor (*p* = 0.0019) and having a white-collar job was a facilitating factor (*p* = 0.0185) for participants with early RTW who were still working at the 12-month follow-up. Furthermore, Table 4 shows a comparison between the latter group and the participants with late RTW. Stage III–IV (*p* = 0.0169) was the only hindering factor that differed between the two groups of participants and no other significant differences between the two groups were observed (*p* = 0.0169).

In Table 4, the left panel shows the results of a comparison between the participants who returned to work (RTW) at some point within 3 months after treatment and continued to work at the 12-month follow-up and those who did not RTW (did not RTW within 3 months or could not continue at 12-month follow-up). The right panel shows the results of a comparison between the former group (early RTW and continuing working) and the participants with a late RTW.

The first column is the action variables. The second column (Yes) reports the percentage numbers of RTW (RTW rate) exposed to the action variable, and the parenthesis is the total number of participants in the action group. The third column (No) reports the percentage numbers of RTW not exposed to the action variable, and the parenthesis is the total number of patients in the control group. The fourth column (ACEe) reports the estimates of the ACE. Intuitively, it is supposed to be the difference between the percentages in the second and third columns. However, as we employed the propensity score matching (PSM) technique in this study, the action and control groups have been rebalanced according to the estimated propensity scores. We observe that the estimate of ACE in the second row, for example, is −0.1910 which even has different sign compared to the difference between 52.54% and 45.79% This also reflects the fact that the original two groups are not balanced and there is confounding. The fifth column (ACEp) reports the corresponding p-values of the estimated ACE. If the estimate is significant, a positive value implies that it facilitates RTW, whereas a negative value implies that it hinders RTW. If the estimate is significant, a positive value implies that it facilitates RTW, whereas a negative value implies that it hinders RTW. Excluded participants are those who retired or are deceased.

## 4. Discussion

In this prospective multicentre observational study on participants with HNC, the clinical factor found to hinder early RTW was advanced tumour stage, and oral and larynx cancer hindered late RTW. Moreover, participants with more physically demanding jobs were less likely to RTW. In total, 227 participants with an age of 65 years or less at the initiation of treatment were assessed for RTW one year after the end of treatment; of them, 92 participants (41%) had early RTW. Among the 131 (58%) participants working 12 months after the end of treatment, 126 participants were tumour free, and five participants still suffered from cancer.

The percentage of participants’ RTW and not RTW outcomes in this study differed according to the site and stage of HNC. We have, in an earlier study, showed that 72% of 295 individuals with oropharyngeal cancer were working 15 months after diagnosis [24]. Other studies have shown that in individuals with oral cancer (*n* = 174), 55% had RTW at a follow-up of 6 months or more after the termination of treatment [25], and in a study of 111 individuals with different HNC diagnoses, 44.1% had RTW within 5 years [26]. In a review of follow-ups of HNC survivors, the rate of RTW varied between 32 and 90%, 3.6–11 months after the end of treatment [3]. The wide RTW range in that study may be explained by the heterogeneous nature of HNC, which agrees with the importance of disease status and treatment revealed in the present study. In addition, HNC survivors have a complex burden of unresolved physical, psychological, and existential needs that add to their risk profile for not returning to work [6].

The recurrence rate in patients with HNC is highest during the two first years after initial treatment [27]. The present study included all participants enrolled at baseline, and not only patients who remained cancer-free during the disease trajectory, as the study aimed to prospectively describe a real-world situation for cancer survivors. The RTW is not the same as that of a cancer-free individual. Of the 131 cancer survivors working at the 12-month follow-up, 9 had cancer recurrence after completion of treatment, and 5 of them were still not cancer-free even though they were working.

To better understand the time to RTW pattern, participants were followed-up at 3 and 12 months after the end of treatment. The results showed a significant difference between participants with cancer stage I-II and III-IV regarding working and being on sick leave at both the 3-month follow-up and at the 12-month follow-up. Additionally, the advanced stage (III-IV) significantly hindered the participants from RTW and a similar association with RTW has been demonstrated among earlier studies [26,28].

The pattern of RTW was not shown to be robust or predictable, even in patients with early RTW, as periodic RTW was observed for medical and social reasons. Although 80 of 92 (87%) participants who had RTW at three months still were working nine months later, 12 participants practised periodic RTW and had left the work sector due to retirement, sick leave, drop out, or were deceased. The inability to RTW or to discontinue working after cancer treatment was impacted by multiple factors: among the 43 participants in this cohort, the sick leave period ended in retirement at any time during the study period, and 26 participants were on sick leave throughout the entire follow-up.

To examine RTW patterns, appropriate facilitators and hindrances were split into three categories and used as predictors: clinical (tumour site and tumour stage), work-related (white-, blue- and pink-collar workers), and sociodemographic factors (age, educational status, marital status, living condition and smokers). The oral cavity, oropharynx, and larynx are the three most common anatomical sites for head and neck squamous cell cancers. Although HNC affects individuals of different ages, occupations, and living conditions [4], the most important finding in the present study was that clinical factors affected RTW ability. This finding was not unexpected, as the study cohort displayed cancers with different HNC sites, stages, and treatments. Patients with advanced-stage disease were more prone to recurrence and cancer-related death, which agrees with the finding that advanced-stage disease is a negative factor for RTW. Individuals treated for oral cavity and laryngeal cancers had a lower rate of RTW. Considering the different variables affecting RTW, it is difficult to construct a comprehensive model for predicting RTW in a mixed population of patients with HNC.

Psychosocial and physical demands are important aspects of the workload and RTW after HNC treatment. In the present study, the individuals were divided into three occupational categories white-, blue- and pink-collar workers. The results show that white-collar work facilitates RTW. In another study of 80 patients with HNC, including more than six months after the latest treatment, pink-collar work was significantly associated with no RTW [28].

RTW is important for many cancer survivors as it symbolises a regaining of normality and daily life [29], where work represents a meaningful aspect of life [30]. Support plays an important role in RTW, and in the present study, support from a spouse or partner was demonstrated to be most likely a facilitating factor for late RTW, which is in agreement with the results of a systematic review of cancer survivors in Europe with different cancer diagnoses, including HNC [31]. However, international comparisons of RTW are complex as several factors, such as the work environment and policy, cultural contexts, and economic issues, may affect RTW patterns [32]. Comparative approaches must also consider that different countries have different social security systems [33]. In the Swedish system [34], employers pay sick pay for the first 1–14 days. Subsequently, the person on sick leave must thereafter apply for sickness benefits through the Swedish Social Insurance Agency. If an employee is assumed to be sick for more than 60 days, a rehabilitation plan needs to be implemented by the employer for easier RTW [34]. The support from the employer and the obligation to give support with a rehabilitation plan can be an important factor for an individual with HNC to RTW [29]. The reduction in money when you are on sick leave may also directly impact the requirement for a person to RTW as soon as possible, or even if you are not on sick leave at all.

Healthcare services also have an important task of identifying rehabilitation needs and using experienced rehabilitation staff to effectively train patients after treatment for HNC. Even though rehabilitation should be available for all HNC survivors, the findings of the present study support that extra attention should be given to patients with advanced-stage cancer, oral, and larynx cancer and patients living alone. Rehabilitation and screening for psychological and physical impairments are important for patients with cancer to preserve function and improve quality of life [35]. Efforts from different rehabilitation competencies such as occupational therapists, dietitians, physiotherapists, counsellors, speech therapists, and psychologists are often needed. Interdisciplinary rehabilitation programs are still lacking in many HNC centres [36]. Professionals must work together to address the complex symptoms and problems that can arise in patients with HNC [37].

Regarding the dearth of sociodemographic factors, further studies should address the economic burden of a person on sick leave and the impact it might have on RTW and also focus on providing patient rehabilitation for a better chance of preparedness and success in RTW.

The limitations of this study are the rather short follow-up time and the inclusion of participants with a WHO performance 0–2, which means that patients with a less favourable status were excluded from the study. A further limitation is the lack of patient-reported outcome measures and that data were not collected regarding income.

## 5. Conclusions

The majority of participants working at the 3-month follow-up were still working at the 12-month follow-up. A clear RTW pattern was observed. Factors hindering RTW included advanced tumour stage and oral and larynx cancer. White-collar workers and participants living with spouses or partners were more likely to RTW. Further studies should address the economic burden of a person taking sick leave and its impact on RTW. Healthcare should focus on providing rehabilitation for a better chance and preparedness for success in RTW.

## Figures and Tables

**Figure 1 cancers-15-04705-f001:**
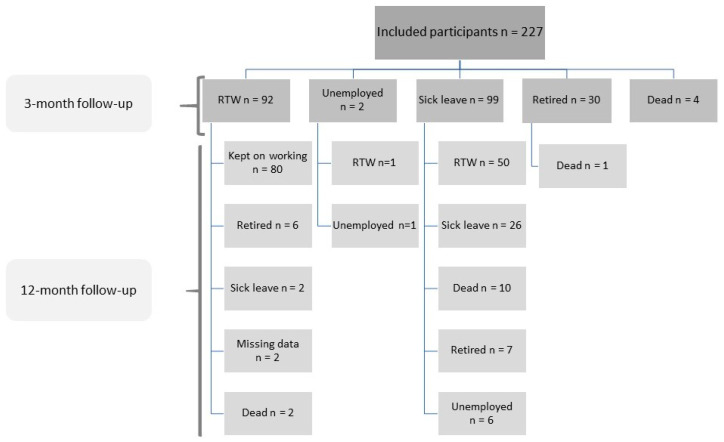
A flow chart of returning to work (RTW) at 3 (early RTW) and 12 months (late RTW) after the end of treatment for 227 study participants with HNC. *n* = numbers are given. HNC: head and neck cancer.

**Table 1 cancers-15-04705-t001:** Clinical, work-related, and sociodemographic characteristics of 227 participants with head and neck cancer (*n* = numbers are given).

Characteristics	Sub-Groups	*n* (%)
Age, mean years (±SD)		55.1 (±8.65)
Age, range of years		22–65
Age	<60	133
	≥60	94
Sex	Female	66 (29.1)
	Male	161 (70.9)
Marital status	Married, cohabiting	170 (74.9)
	Single or couple not living together	57 (25.1)
Living conditions	House	152 (67)
	Owned apartment	22 (9.7)
	Rental flat	53 (23.3)
Educational status	Mandatory	33 (14.5)
	High school/college	119 (52.4)
	Other post-high school education	7 (3.1)
	University	68 (30.0)
Type of work	White collar	112 (49.3)
	Blue collar	87 (38.3)
	Pink collar	25 (11.0)
	Unemployed	2 (0.9)
	Student	1 (0.4)
Tumour site	Oropharynx	113 (49.8)
	Oral cavity	60 (26.4)
	Larynx	19 (8.4)
	Nasopharynx	12 (5.3)
	Cancer of unknown primary	10 (4.4)
	Nasal and sinus	7 (3.1)
	Salivary glands	3 (1.3)
	Hypopharynx	2 (0.9)
	Other ^1^	1 (0.4)
Tumour stage UICC ^2^ 8	I	93 (41.1)
	II	52 (22.9)
	III	42 (18.5)
	IV	39 (17.2)
	Not applicable	1 (0.4)
Treatment type	Surgery	20 (8.8)
	Radiotherapy (RT)	83 (36.6)
	Chemo ^3^ radiotherapy (CRT)	63 (27.8)
	Surgery and RT or CRT	61 (26.9)

^1^ Rhabdomyosarcoma in the left maxilla ethmoidal. ^2^ The Union for International Cancer Control’s (UICC). ^3^ Cisplatin or Cetuximab.

**Table 2 cancers-15-04705-t002:** Return to work or on sick leave status of participants treated for head and neck cancer at the 3- and 12-month follow-up categorised by cancer stage (I–II and III–IV).

Follow-Up	Stage *n* (%)	Sick Leave *n* (%)	Working *n* (%)	Pearson’ Chi-Squared Test
3-month ^1^	I and II	53 (54.1)	72 (78.3)	*X*^2^ = 12.326 df = 1 *p* = 0.001
	125 (65.8)			
	III and IV	45 (45.9)	20 (21.7)	
	65 (34.2)			
12-month	I and II	11 (39.3)	93 (71.0)	*X*^2^ = 10.250 df = 1 *p* = 0.001
	104 (65.8)			
	III and IV	17 (60.7)	38 (29.0)	
	65 (34.2)			

^1^ Stage is not applicable in one participant being on sick leave at the 3-month follow-up.

**Table 3 cancers-15-04705-t003:** Factors influencing return to work (RTW) at 3 and 12 months after treatment completion for head and neck cancer.

	RTW 3-Month Follow-Up				RTW 12-Month Follow-Up			
Action Variables	Yes	No	ACEe	ACEp	Yes	No	ACEe	ACEp
Age (≥60)	42.86% (63)	50% (130)	−0.0242	0.7790	78.72% (47)	78.99% (119)	0.0442	0.6327
University or college education	53.85% (65)	44.53% (128)	−0.1047	0.4565	84.75% (59)	75.7% (107)	−0.1353	0.3212
Living in a relationship	47.55% (143)	48% (50)	0.0199	0.8890	83.06% (124)	66.67% (42)	0.2402	0.0445
Living in a house	51.56% (128)	40% (65)	0.0807	0.4333	82.88% (111)	70.91% (55)	0.0041	0.9615
White-Collar	52.58% (97)	42.71% (96)	0.0819	0.5318	83.53% (85)	74.07% (81)	0.2994	0.0267
Pink-Collar	40.91% (22)	48.54% (171)	−0.3497	0.1445	70% (20)	80.14% (146)	−0.0151	0.9474
Oropharynx	46.32% (95)	48.98% (98)	0.1641	0.4142	84.88% (86)	72.5% (80)	0.0753	0.5684
Oral	50% (52)	46.81% (141)	−0.3333	0.1767	70% (40)	81.75% (126)	−0.5693	0.0210
Larynx	31.25% (16)	49.15% (177)	0.3575	0.3354	57.14% (14)	80.92% (152)	−0.7530	0.0041
Advanced cancer stage (III or IV)	29.85% (67)	57.14% (126)	−0.3359	0.0038	67.86% (56)	84.55% (110)	−0.0934	0.3654
Smoking	44.12% (102)	51.65% (91)	0.0108	0.9022	77.01% (87)	81.01% (79)	−0.0261	0.7509

**Table 4 cancers-15-04705-t004:** Returning to work after head and neck cancer.

	The Results for RTW 3–12 m				Early RTW and Continuing Working against Late RTW			
Action Variables	Yes	No	ACEe	ACEp	Yes	No	ACEe	ACEp
Age (≥60)	44.68% (47)	49.58% (119)	0.0528	0.6406	56.76% (37)	62.77% (94)	0.0853	0.4639
University or college education	52.54% (59)	45.79% (107)	−0.1910	0.1321	62% (50)	60.49% (81)	−0.2883	0.0982
Living in a relationship	47.58% (124)	50% (42)	0.0392	0.7717	57.28% (103)	75% (28)	−0.2672	0.0976
Living in a house	51.35% (111)	41.82% (55)	0.1687	0.0967	61.96% (92)	58.97% (39)	0.1595	0.2378
White-Collar	55.29% (85)	40.74% (81)	0.3301	0.0185	66.2% (71)	55% (60)	0.3270	0.0568
Pink-Collar	35% (20)	50% (146)	−0.3373	0.1862	50% (14)	62.39% (117)	−0.4771	0.1443
Oropharynx	46.51% (86)	50% (80)	0.1130	0.5673	54.79% (73)	68.97% (58)	−0.0407	0.8328
Oral	55% (40)	46.03% (126)	−0.2199	0.4373	78.57% (28)	56.31% (103)	0.3817	0.1279
Larynx	35.71% (14)	49.34% (152)	−0.4578	0.1586	62.5% (8)	60.98% (123)	0.3206	0.3671
Advanced cancer stage (III or IV)	32.14% (56)	56.36% (110)	−0.3579	0.0019	47.37% (38)	66.67% (93)	−0.3359	0.0169
Smoking	43.68% (87)	53.16% (79)	−0.0381	0.6698	56.72% (67)	65.62% (64)	−0.0623	0.5588

## Data Availability

Data are maintained in this article. Data are not publicly available due to privacy.

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
