# Peer review of "Predicting Return to Work after Head and Neck Cancer Treatment Is Challenging Due to Factors That Affect Work Ability"

_cancers, 2023, doi:10.3390/cancers15194705_

Round 1

Reviewer 1 Report

 This is a very interesting article about an important aspect of the functioning of patients with head and neck cancer. This is a rarely discussed aspect of the functioning of oncological patients. I congratulate the authors on the topic, especially since they also included advanced stages of cancer. The presentation of the results is clear and transparent, and the discussion is carried out with the papers of recent years. I accept paper in this form.

Author Response

Thank you very much for taking the time to review this manuscript and thank you for your kind comments about a topic we find essential.

Reviewer 2 Report

Thank you for the opportunity to review the manuscript.

Introduction: This section is way too short and ends abruptly. The introduction needs more references and literature, pertaining to what was done in the recent past, with addition of some recent comparison studies and what was found in those studies. The literature does not show sufficient evidence on why the authors chose to examine RTW at intervals of 3 and 12 months after treatment.

Method: What was the rationale for not including 'age' as a variable in the propensity score matching model.

Results: In the introduction section the authors mention that HNC are increasingly common in the young population (line 63). The mean age of participants was 55.1 years (line 173), whereas in table 3 age is included as an action variable with a cut off >=60 years, if the mean age was 55 years, why was a higher cut off included. If the authors think that Age could be an influencing factor for RTW, then why were the participants not matched on it including a few other demographic variables, which could also influence the outcome, such as income.

How were the variables chosen as influencing factors, was this based on a priori literature, if so please specify. Fo example, type of treatment could also be an influencing factor for RTW, why was it no included as an action variable. 

Author Response

Please see the uploaded PDF-file

Reviewer 3 Report

Please review comments inside the attached document as there are clarifications required.

Author Response

Please see the uploaded PDF-file

Reviewer 4 Report

This paper was generally well designed. Its methods were clear, statistical analysis were clear, and the writing was good in general. However, it still needs some major edits to provide more clarity, the definitions of some key concepts in this paper need more supportive materials and evidence, and the main findings of this paper were not so informative for the readers. The evidence to support the disease stage, work type and having a spouse or partner as factors influencing RTW among cancer survivors could been easily found in some other similar papers,  so this paper should explain more about the differences of these associated factors between HNC survivors and other cancer types.

 Below, please see some concrete comments and suggestions.

1. Line 51-line 53: Explain the logic of putting the brief introduction about treatments of HNC between the introduction of  the increasing incidence of HNC in young population and the introduction of associated factors for RTW.

2. Line 79- line 82: In the study design part, please provide more evidence about the reason why you choose 3 and 12 months after the end of treatment as the time points.

3. Line 93- line 95: Please explain why you defined the RTW as the following two categories:  early RTW (0-3 months after the end of 93 treatment) and late RTW (3-12 months after the end of treatment). If possible, please provide the references as the definitions of three occupation groups.

4. Line 274 – line 276: Please provide the results of PSM in the supplementary materials.

5. Line 327 line 329: The results of these facilitators and hindrances can't be seen in the results part, so please provide the evidence to support the classification standard.

6. Line 362- line 370: In the results part, there are no results about the rehabilitation and screening for psychological and physical impairments. If the authors want to discuss them, related results should be provided.

7. Line 375 – line 378: Please enrich the limitation part.

8. Line 382 – line384: Please discuss potential implications and recommendations for healthcare professionals, policy makers, and the broader system based on the study findings.

 Minor editing of English language required.

Author Response

Please see the uploaded PDF-file

Round 2

Reviewer 4 Report

I am satisfied with the authors' revisions and feedbacks. And I have no further suggestion.